# Enhanced Performance of GaAs Metal-Oxide-Semiconductor Capacitors Using a TaON/GeON Dual Interlayer

**DOI:** 10.3390/nano13192673

**Published:** 2023-09-29

**Authors:** Lu Liu, Wanyu Li, Fei Li, Jingping Xu

**Affiliations:** School of Integrated Circuits, Huazhong University of Science and Technology, Wuhan 430074, China; liulu@hust.edu.cn (L.L.); wanyuli_134@163.com (W.L.); fei1682022@163.com (F.L.)

**Keywords:** GaAs MOS devices, dual interlayer, TaON/GeON, interface-state density

## Abstract

In this work, a dual interfacial passivation layer (IPL) consisting of TaON/GeON is implemented in GaAs metal-oxide-semiconductor (MOS) capacitors with ZrTaON as a high-k layer to obtain superior interfacial and electrical properties. As compared to the samples with only GeON IPL or no IPL, the sample with the dual IPL of TaON/GeON exhibits the best performance: low interface-state density (1.31 × 10^12^ cm^−2^ eV^−1^), small gate leakage current density (1.62 × 10^−5^ A cm^−2^ at *V_fb_* + 1 V) and large equivalent dielectric constant (18.0). These exceptional results can be attributed to the effective blocking action of the TaON/GeON dual IPL. It efficiently prevents the out-diffusion of Ga/As atoms and the in-diffusion of oxygen, thereby safeguarding the gate stack against degradation. Additionally, the insertion of the thin TaON layer successfully hinders the interdiffusion of Zr/Ge atoms, thus averting any reaction between Zr and Ge. Consequently, the occurrence of defects in the gate stack and at/near the GaAs surface is significantly reduced.

## 1. Introduction

Recently, as Si-based CMOS technology is approaching its fundamental limit, high- mobility semiconductors such as Ge or Ⅲ-Ⅴ compound semiconductors have become inevitable for future CMOS technology development [1,2]. Among them, the GaAs-based MOS device has been considered as the most viable candidate due to its higher mobility, higher breakdown voltage, and larger band gap than Si [3]. Meanwhile, many high-k materials, e.g., HfO_2_ [4], TiO_2_ [5], La_2_O_3_ [6], Y_2_O_3_ [7], etc., have been introduced into GaAs-based MOS devices for the scaling down of the device dimension. However, since GaAs easily forms oxides of Ga and As that create interface defects and contain border traps, high interface-state density (*D_it_*) at the interface of high-k/GaAs is generated, leading to the degradation of device performance [8]. In order to passivate the surface of GaAs, various wet chemical surface treatments have been studied. Sulfide passivation using (NH_4_)_2_S, for example, has been proven to be effective in removing the native oxide and elemental arsenic from the surface of GaAs by creating an S termination. But this termination tends to be unstable when exposed to air or water and thus it has to be protected by a dielectric layer [9]. Additionally, prior research has confirmed that the intrinsic oxides on the GaAs surface can be removed effectively using silicon nitride (Si_x_N_y_) or Al_2_O_3_. However, these methods inevitably lead to a low-k interfacial layer between the GaAs substrate and the high-k gate dielectric, which are unfavorable for decreasing the equivalent oxide thickness (EOT) and improving the dielectric constant [10,11]. Among these high-k materials, Zirconium dioxide (ZrO_2_) has a high *k* value (~24) and a large bandgap (5.8 eV) [12], and has been used as a high-k gate dielectric for GaAs MOS devices with good electrical properties [13,14,15]. However, the low crystallization temperature (400–500 °C) of ZrO_2_ may facilitate its crystallization [16,17], thereby increasing the gate leakage current [17,18]. Fortunately, doping Ta into ZrO_2_ can improve its crystallinity, and the high *k* value (~25) of Ta oxide can maintain a high dielectric constant for the novel binary oxide [19]. In addition, incorporating nitrogen into the oxide can usually increase the *k* value, reduce gate leakage current, passivate oxygen vacancies, and form strong N-related bonds at/near the dielectric/semiconductor interface to further enhance the thermal stability and reliability of the devices [20,21]. Therefore, in this work, Ta-doped Zr-oxynitride (ZrTaON) will be employed as a high-k gate dielectric to fabricate GaAs MOS devices. Further, in order to improve the interfacial properties of the devices, an interfacial passivation layer (IPL) is typically implemented prior to depositing the high-k layer [6,14,20]. Due to the excellent crystallographic matching between germanium (Ge) and GaAs (the lattice constant is 5.653 Å for GaAs and 5.658 Å for Ge [22]) and the passivation effect on the defect-related Ga/As-O or As-As bonds, Ge could be regarded as a promising IPL material for GaAs MOS devices. However, Ge is an amphoteric dopant for GaAs, which may alter the doping concentration or even induce the counterdoping of the GaAs substrate when using it as IPL [23]. So, Ge oxynitride (GeON) will be used as an IPL instead of Ge. Furthermore, to avoid the reaction between Zr (from ZrTaON) and Ge (from GeON) that can degrade the quality of the gate stack [24], a thin TaON film, which can strongly block interdiffusion of elements, is inserted between the GeON IPL and high-k layer. As a result, the GaAs MOS capacitor with ZrTaON as the high-k gate dielectric and TaON/GeON as the dual IPL is proposed and fabricated, and excellent interfacial and electrical properties are achieved for the device.

## 2. Experiments

N-GaAs (100) wafers with a resistivity of 0.002–0.005 Ωcm were employed to fabricate the MOS capacitors. The wafers were firstly degreased in acetone, ethanol, and isopropanol, and then dipped in diluted HCl to remove the native oxides. This was followed by (NH_4_)_2_S dipping for 40 min at room temperature for surface sulfur passivation, and finally rinsing in de-ionized water for several times [25]. After drying with N_2_, the wafers were divided into different samples that were fabricated with different gate stacks: (1) the sample would receive ~1 nm GeN and then ~1 nm TaN as the dual IPL by sputtering Ge (RF) and Ta (DC) targets, respectively, followed by depositing ~8 nm ZrTaN as the high-k layer via co-sputtering Zr (RF) and Ta (DC) targets (TaON/GeON sample); (2) the sample would receive ~2 nm GeN (IPL) and then ~8 nm ZrTaN (high-k layer) (GeON sample); (3) the sample would only receive ~10 nm ZrTaN film as the high-k layer (control sample). The cross-section diagrams of the three samples are shown in Figure 1, respectively. The sputtering was performed in an ambient of Ar:N_2_ = 24 sccm: 12 sccm at room temperature (RT) using a Denton Vacuum Discovery Deposition System. Subsequently, these samples were annealed at 600 °C for 60 s in an atmosphere of N_2_ (500 sccm) + O_2_ (50 sccm) to form the ultimate oxynitrides. Finally, Al was thermally evaporated and patterned by photolithography as the gate electrode (7.85 × 10^−5^ cm^2^) and also as back electrode, followed by forming-gas (5% H_2_ + 95% N_2_) annealing at 300 °C for 20 min to reduce the contact resistance [25].

The capacitance–voltage (C-V) and *J_g_-V_g_* curves (*J_g_* is the gate leakage current density and *V_g_* is the gate voltage) were measured using an Agilent 4284A precision LCR meter and a Keithley 4200-SCS semiconductor characterization system from the United States, respectively. The XPS method was used to analyze chemical states at/near the gate dielectric/GaAs interface using a XPS equipment with a type of VG Multilab 2000 produced by Thermo Fisher in the United States. The physical thickness of the gate dielectric was determined by spectroscopic ellipsometry produced by J. A. Woollam in the United States. All measurements were performed in a dark environment with electromagnetic shielding at RT.

## 3. Results and Discussion

Figure 2 shows the typical HF (1-MHz) C-V curves of the three samples, swept in two directions (from inversion to accumulation and back). It is evident that compared to the control sample, the GeON sample and especially the TaON/GeON sample display significantly improved electrical characteristics with reduced “stretch-out” and a more saturated accumulation region, indicating an improved high-k/GaAs interface with fewer defect-related Ga/As-O and As-As bonds (as confirmed by the XPS analyses below) [6,20]. Also, the accumulation capacitance for the TaON/GeON and GeON samples is much larger than that of the control sample, which should be attributed to the reduced low-k Ga/As oxides on the GaAs surface for the two samples [6]. The largest accumulation capacitance for the TaON/GeON sample can be attributed to the least interfacial Ga/As oxides and the much larger *k* value of TaON compared to GeON, which results in the largest equivalent *k* value (18) and the smallest capacitance equivalent thickness (*CET* = *ε_0_k_SiO_*_2_/*C_ox_* = 2.17 nm, where *k_SiO_*_2_ is the relative permittivity of SiO_2_) for the stacked gate dielectric, as summarized in Table 1. The hysteresis voltages (Δ*V_fb_*) obtained from the HF C-V measurements are found to be 60 mV, 70 mV, and 130 mV for the TaON/GeON, GeON, and control samples, respectively. These results suggest that the TaON/GeON sample exhibits the least pronounced slow states, primarily located in the IPL and near/at the IPL/GaAs interface. The enhanced electrical properties observed in the GeON sample can be attributed to the inhibitory role of the GeON IPL, which prevents the out-diffusion of Ga/As atoms and the in-diffusion of oxygen. Consequently, the GaAs surface is protected from oxidation, preserving the quality of the interface. Furthermore, the insertion of a 1 nm TaON layer between the high-k layer of ZrTaON and the GeON IPL is proved to be effective in preventing the reaction between Zr and Ge by blocking the interdiffusion of Zr/Ge atoms [24]. This additional layer contributes to the maintenance of the good quality of the GeON IPL and further reduces defects near/at the GeON/GaAs interface. As a result, it enables the achievement of optimal interfacial properties for the TaON/GeON sample.

Some other physical and electrical parameters of the capacitors can also be extracted from the HF C-V curves of the three samples, like oxide capacitance on unit area (*C_ox_*), flat-band voltage (*V_fb_*), equivalent density of oxide charges (*Q_ox_*), equivalent gate dielectric *k* value (*k* = *C_ox_*·*T_ox_*/*ε*_0_; *T_ox_* is the total gate dielectric thickness; *ε*_0_ is the vacuum dielectric constant), and density of interface states (*D_it_*) evaluated by Terman’s method [26], as listed in Table 1. The samples with IPL exhibit much lower *D_it_* than the sample without IPL (the control sample), implying a great passivation effect of the GeON IPL, especially the TaON/GeON dual IPL. Also, the positive *V_fb_* shows the presence of negative oxide charge (*Q_ox_*) in the gate stack, and the smallest *V_fb_* (0.70 V) for the TaON/GeON sample indicates the least *Q_ox_* in the gate stack and thus the best quality of the gate dielectric, due to less Ga/As out-diffusion and a reduced reaction between Zr and Ge, as mentioned above.

Figure 3 shows the frequency dispersion of the three samples measured at different frequencies. Clearly, the control and GeON samples show a large dispersion behavior in both the depletion and accumulation regions, which is probably due to the effects of the interface traps [27,28] and the border traps in the gate dielectric near the interface [29], respectively. Fortunately, using the dual IPL of TaON/GeON can effectively prevent the reaction between Zr and Ge to maintain the good quality of the GeON IPL and achieve an excellent passivation effect on the GaAs surface, thus largely reducing the interface traps and leading to much improved frequency dispersion properties, as shown in Figure 3.

The gate leakage properties (*J_g_* vs. *V_g_*) of the three samples are shown in Figure 4. The control sample exhibits much worse gate leakage properties than the GeON and especially TaON/GeON samples; at *V_fb_* + 1 V, the gate leakage current is 5.88 × 10^−3^ A/cm^2^, 4.39 × 10^−5^ A/cm^2^, and 2.23 × 10^−5^ A/cm^2^ for the control, GeON, and GeON/TaON samples, respectively. The large gate leakage current for the control sample should originate from the strong trap-assisted tunneling of carriers by the traps in the gate dielectric and at the high-k/GaAs interface, associated with its large *Q_ox_* and *D_it_* values, as listed in Table 1. Also, the reduction of the conduction-band offset induced by the much Ga/As oxides at the interface may be another reason [27]. For the TaON/GeON sample, however, the smallest gate leakage current is obtained due to the largely improved gate stack quality and interface properties by the dual IPL of TaON/GeON.

To confirm the above discussion, XPS analyses were carried out for the three samples to investigate the interfacial chemical states between the high-k dielectric and GaAs substrate. For this purpose, the thickness of the gate dielectric layer is thinned to ~3 nm from the GaAs surface by using an in situ Ar^+^ ion beam. The energy scale of the three samples is calibrated by fixing the C 1s at a binding energy (BE) of 285 eV to eliminate the charging effect. The Zr 3d spectra for the three samples are shown in Figure 5, indicating the presence of Zr in the gate stack. Also, the presence of N and O can be confirmed by the O 1s and N 1s spectra in the insets of Figure 5. In Figure 6, showing the Ta 4f spectra for the three samples, the two peaks at 26.51 eV and 28.36 eV originated from Ta_2_O_5_, respectively, and the two peaks at 24.65 V and 26.50 eV should originate from TaO_2_ [30], demonstrating the existence of Ta oxides in the IPL and near the GaAs surface. These results confirm the formation of the ZrTaON high-k layer. Obviously, in Figure 6, the peak intensity of TaO_2_ for the TaON/GeON sample is stronger than that for the other two samples due to presence of the 1 nm TaON interlayer. Especially, in Figure 6, a Ge 3d peak can be found for the TaON/GeON and GeON samples, indicating the presence of Ge on the GaAs surface. The stronger Ge 3d peak for the GeON sample than the TaON/GeON sample should be due to the thicker GeON IPL for the former (~2 nm) compared to the latter (~1 nm). In Figure 6a, showing the TaON/GeON sample, the Ge 3d peak is located at 30.25 eV, which should be assigned to GeO_x_N_y_ [31]; however, the Ge 3d peak in Figure 6b, showing the GeON sample, has a positive shift of ~0.85 eV. Similarly, the Zr 3d spectrum for the GeON sample in Figure 5 also has a slight shift to higher energy as compared to the other two samples. These are associated with the reaction between Zr and Ge in the gate stack to form ZrGeON [24], which would degrade the gate stack quality and thus the electrical properties of the devices. However, the reaction between Zr and Ge has been greatly suppressed for the TaON/GeON sample, which can be attributed to the strong blocking role of the 1 nm TaON IPL against the interdiffusion of the Zr and Ge atoms, thus avoiding the degradation of the gate stack and maintaining the excellent interface quality and electrical properties.

The high *D_it_* observed on the GaAs surface can primarily be attributed to the presence of native oxides of Ga and As. Additionally, another potential source of *D_it_* is arsenic, which can exist in the form of dimers or as elemental arsenic. Arsenic dimers are commonly found in various surface reconstructions of GaAs, whereas elemental arsenic may result from preferential oxidation of Ga and the instability of As_2_O_3_ on the GaAs surface, as described by the chemical reaction [10]: As_2_O_3_ + 2GaAs → 4As + Ga_2_O_3_. The As 2p3/2 and Ga 3d spectra of the samples are shown in Figure 7 and Figure 8, respectively, where the As-S, Ga-S, and Ga-N peaks can be found due to the sulfur passivation by the (NH_4_)_2_S dipping and the nitridation induced by sputtering in an N_2_-contained ambient. In Figure 7, an As-O peak can be found at 1326.2 eV for the three samples, and in Figure 8, all three samples exhibit a Ga-O peak at 21.1 eV, indicating the oxidation of the GaAs surface. As compared to the control sample, the As/Ga-O peaks and the As-As peaks are weakened for the GeON sample and especially for TaON/GeON sample, implying that the formation of low-k native oxides on the GaAs surface and arsenic dimers is effectively suppressed by GeON IPL, especially by the dual IPL of TaON/GeON, due to its strong blocking role against O in-diffusion toward the GaAs substrate and the reduced reaction between Zr and Ge in the gate stack. The strong As/Ga-O peaks and As-As peaks of the control sample in its As 2p3/2 and Ga 3d spectra (Figure 7c and Figure 8c) indicate a large amount of low-k Ga/As-oxides and arsenic dimers on the GaAs surface, leading to a much smaller accumulation capacitance in Figure 2 and a high *D_it_*. In addition, the weakest As-As peak is also observed in Figure 7 for the TaON/GeON sample, suggesting the suppressed formation of the As-As bonds on the GaAs surface. It is well known that the Ga/As-O and As-As bonds are closely related to the defects at/near the surface, and so it can be concluded that the use of the TaON/GeON dual IPL can effectively reduce the defects at/near the GaAs surface, thus leading to much improved electrical properties of the devices, as described above.

A comprehensive comparison of the electrical and physical properties between the samples presented in this study and other existing solutions in the literature is provided in Table 2. The results clearly demonstrate that this work yields the smallest flat-band voltage and a *D_it_* value that is comparable to, and in some cases even lower than, those reported in the literature. Notably, these superior outcomes are particularly pronounced for the samples exhibiting high *k* values (exceeding 18).

## 4. Conclusions

The GaAs MOS capacitors with GeON IPL or TaON/GeON dual IPL are fabricated and their interfacial and electrical properties are investigated. A comparison is made with a control sample lacking any IPL. The results demonstrate that the inclusion of the GeON IPL, and particularly the TaON/GeON dual IPL, effectively impedes the out-diffusion of Ga/As atoms and the in-diffusion of oxygen. Consequently, it provides protection against oxidation of the GaAs surface. Inserting a thin TaON layer between the high-k ZrTaON layer and the GeON IPL is proved to be even more effective in blocking Ga/As out-diffusion and oxygen in-diffusion. Simultaneously, it efficiently suppresses the reaction between Zr and Ge by preventing the interdiffusion of Zr/Ge atoms. As a result, defects near/at the GaAs surface are further reduced, preserving the quality of the stacked gate and avoiding its degradation. Therefore, it can be concluded that the utilization of the TaON/GeON dual IPL can lead to improved gate stack quality and interfacial properties. Consequently, it yields excellent electrical properties for GaAs MOS devices.

## Figures and Tables

**Figure 1 nanomaterials-13-02673-f001:**
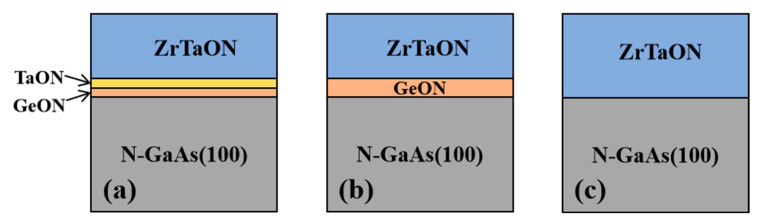
The cross-section diagrams for the three samples: (**a**) TaON/GeON sample, (**b**) GeON sample, and (**c**) control sample.

**Figure 2 nanomaterials-13-02673-f002:**
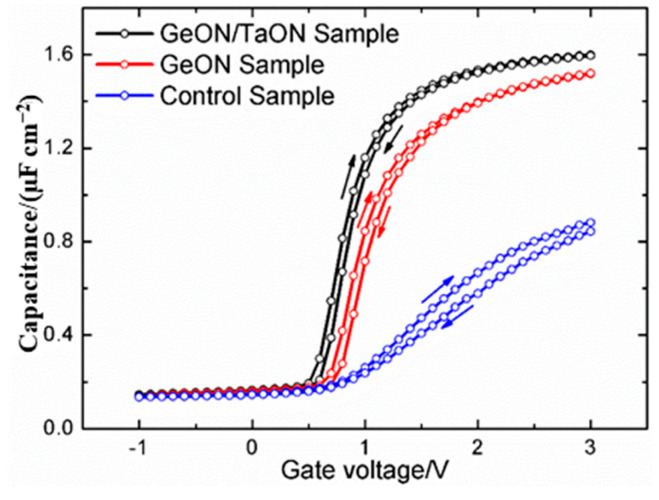
HF (1-MHz) C-V curves for the three samples.

**Figure 3 nanomaterials-13-02673-f003:**
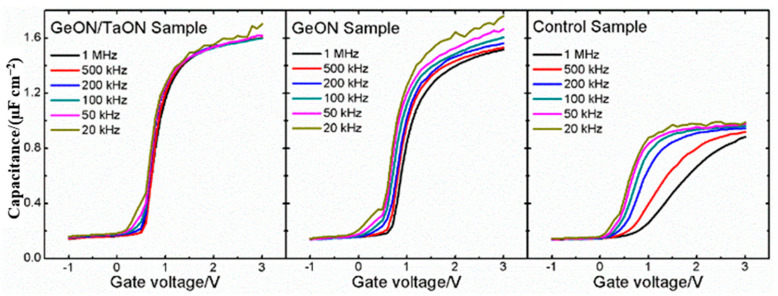
Frequency dispersions of C-V curves at room temperature for the three samples.

**Figure 4 nanomaterials-13-02673-f004:**
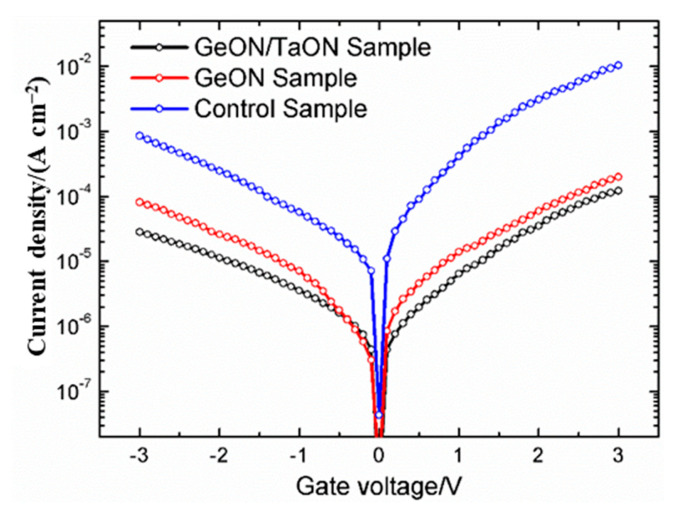
*J_g_* vs. *V_g_* characteristics for the three samples.

**Figure 5 nanomaterials-13-02673-f005:**
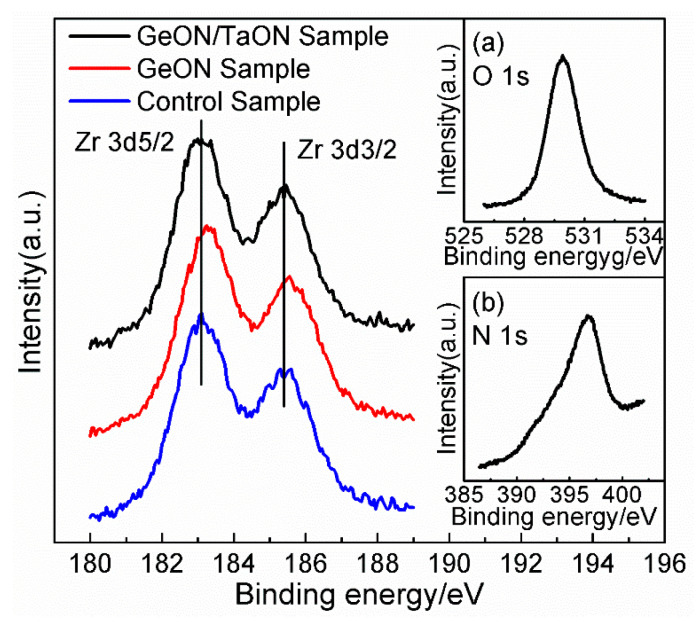
XPS spectrum of Zr 3d for the three samples. The insets (**a**,**b**) are the O 1s and N 1s spectra, respectively.

**Figure 6 nanomaterials-13-02673-f006:**
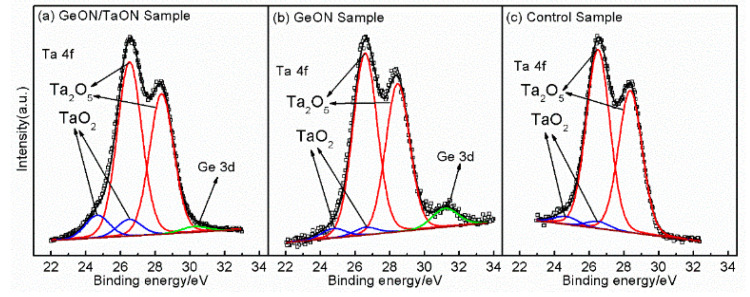
XPS spectra of Ta 4f for the three samples.

**Figure 7 nanomaterials-13-02673-f007:**
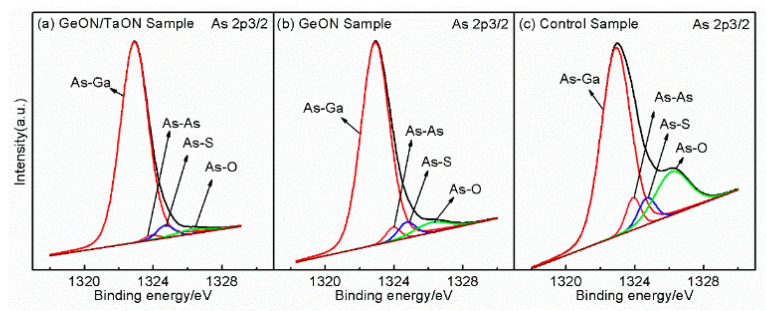
XPS spectra of As 2p3/2 for the three samples.

**Figure 8 nanomaterials-13-02673-f008:**
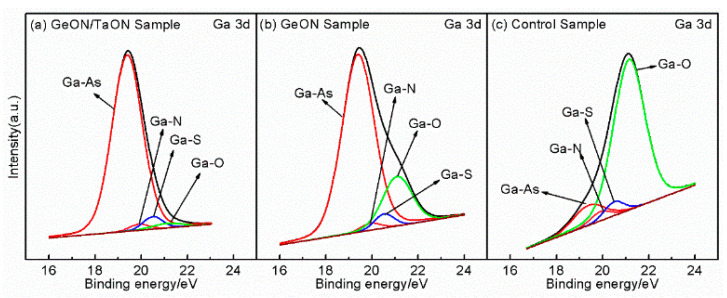
XPS spectra of Ga 3d for the three samples.

**Table 1 nanomaterials-13-02673-t001:** Parameters of the GaAs MOS capacitors extracted from their HF C-V curves.

Samples	*V_fb_* (V)	*D_it_* (/cm^2^eV)	*C_ox_* (μF/cm^2^)	*Q_ox_* (10^12^) (/cm^2^)	*k*	*CET* (nm)
TaON/GeON	0.70	1.31 × 10^12^	1.60	−5.98	18.0	2.17
GeON	0.85	1.65 × 10^12^	1.52	−7.09	17.2	2.27
Control	1.41	8.06 × 10^12^	0.88	−7.17	10.0	3.90

**Table 2 nanomaterials-13-02673-t002:** Comparison of electrical and physical properties of the GaAs MOS devices between this work and other works in the literature.

	Samples	This Work	Y_2_O_3_/Al_2_O_3_ [7]	GGO/HfTiON [20]	AlN/Al_2_O_3_ [22]	LaGeON/ZrON [25]
Parameters						
*k*		18	8.2	25.1	7.9	12.7
*V_fb_* (V)		0.7	~1.0	1.28	~1.0	0.8
*D_it_* (cm^−2^ eV^−1^)		1.3 × 10^12^	2.5 × 10^12^	3.3 × 10^12^	2 × 10^12^	1.2 × 10^12^

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
