# Peer review of "Enhanced Performance of GaAs Metal-Oxide-Semiconductor Capacitors Using a TaON/GeON Dual Interlayer"

_nanomaterials, 2023, doi:10.3390/nano13192673_

Round 1
Reviewer 1 Report
Lu Liu and Jing-ping Xu report in their manuscript entitled “Enhanced performances of GaAs metal-oxide-semiconductor capacitor by using TaON/GeON dual interlayer” about experimental studies on metal oxide semiconductor (MOS) capacitors prepared on GaAs substrates. Although many material properties of GaAs are superior for technological applications, Si is still predominant in data processing and communication technologies. One of the reasons for this is the unfavorably high state density found at interfaces between GaAs and insulators. The present study reports about the fabrication of GaAs-MOS capacitors with a novel dielectric layer design that exhibit low interface state densities and leakage currents coming close to those values found on Si devices. Hence, I evaluate the report of high interest to the readership of the nanomaterials journal.
In more detail, GaAs-MOS capacitors with three different layer designs are studied with capacitance-voltage (CV) as well as leakage current-voltage (IV) measurements and with material analysis employing X-ray photoelectron spectroscopic (XPS). The CV and IV results show convincingly that a design employing an interface passivation layer, which consists of Ge oxynitride and Ta oxynitride between the GaAs substrate and a Ta-doped Zr-oxynitride (TaZrON) as high-k gate dielectric, shows much lower interface-state density and leakage current than reference capacitors prepared with just a ZrTaON dielectric layer or a simpler GeON interface layer between the ZrTaON dielectric and GaAs. The authors discuss their finding in view of the material properties and diffusion behavior of Ge and Ta on GaAs. Finally, the discussion is corroborated by the XPS material analysis.
In summary, the manuscript is of high interest, very well written and reports sound work. I thus can recommend publication in the present form.
Author Response
Thanks very much for the reviewer’s comments.
Reviewer 2 Report
In the manuscript, the Authors present an article about enhanced performances of GaAs metal-oxide-semiconductor capacitors by using TaON/GeON dual interlayer. This issue is very current in the field of electronics.
The article is well-written and organized.
There was a problem with the numbering of individual sections (each section has the same number).
The experiment section lacks photos of samples and their cross-sections as well as in-depth chemical analysis. Attaching these results would make it possible to prove the quality of the samples made.
The presented work also lacks a comparison of the properties of the produced samples with other solutions available in the literature.
I think that the manuscript hasn't sufficient scientific quality and relevance for Nanomaterials in this form. I suggest accepting it after minor revision.
Author Response
In the manuscript, the Authors present an article about enhanced performances of GaAs metal-oxide-semiconductor capacitors by using TaON/GeON dual interlayer. This issue is very current in the field of electronics.
The article is well-written and organized.
- There was a problem with the numbering of individual sections (each section has the same number).
The numbering of individual sections has been changed.
- The experiment section lacks photos of samples and their cross-sections as well as in-depth chemical analysis. Attaching these results would make it possible to prove the quality of the samples made.
The cross-section diagrams for the three samples have been added to show different gate-stacked structures of GaAs MOS. (Figure 1) Some in-depth chemical analysis has been added. (P. 6, L1-6)
- The presented work also lacks a comparison of the properties of the produced samples with other solutions available in the literature.
This is a good recommendation. A comparison on the properties between our samples and other solutions available in the literature have been done, as listed in Table 2. (P. 7, L1-5 and Table 2)
I think that the manuscript hasn't sufficient scientific quality and relevance for Nanomaterials in this form. I suggest accepting it after minor revision.
The relevant revisions have been done to improve scientific quality of our manuscript. (P. 1, L1-19 in introduction; P. 2, L2-6, L12-13, L16-19 and Figure 1 in the experiment section; P. 5, L12-14; P. 6, L1-6; P. 7, L1-5 and Table 2)

Reviewer 3 Report
Researchers describe the enhanced performance of GaAs metal-oxide-semiconductor capacitors using TaON/GeON dual interlayers. To obtain excellent interfacial and electrical properties, they used a dual interfacial passivation layer (IPL) of TaON/GeON implemented for GaAs MOS capacitors. There are a few minor corrections I need to make before publication.
1. There is a need for the author to improve the English in the manuscript.
2. The author should conduct more literature surveys since the introduction is very short.
3. It is necessary for the author to discuss more about experiment devices since he did not mention anything in the manuscript. In this manuscript, he should discuss this more briefly.
4. It is important that the author mentions the value in the figure where he discusses intensity. This axis is misleading since it is blank.
5. The author should discuss his results with recent literature results and describe the table in the discussion.
There is a need for the author to improve the English in the manuscript.
Author Response
Researchers describe the enhanced performance of GaAs metal-oxide-semiconductor capacitors using TaON/GeON dual interlayers. To obtain excellent interfacial and electrical properties, they used a dual interfacial passivation layer (IPL) of TaON/GeON implemented for GaAs MOS capacitors. There are a few minor corrections I need to make before publication.
- There is a need for the author to improve the English in the manuscript.
Have done.
- The author should conduct more literature surveys since the introduction is very short.
Have done and added the relevant statements in the introduction. (P. 1, L1-19 in introduction section)
- It is necessary for the author to discuss more about experiment devices since he did not mention anything in the manuscript. In this manuscript, he should discuss this more briefly.
Have done. (P. 2, L2-6, L12-13 and L16-19 in the experiment section; Figure 1)
- It is important that the author mentions the value in the figure where he discusses intensity. This axis is misleading since it is blank.
Usually, the peak intensity on XPS is arbitrary units (a. u.), as done in many references. The chemical composition, chemical states and chemical bonds of thin films can be analyzed by the peak position, peak shape and peak-area ratio of each characteristic spectral peak in XPS energy spectra which is calibrated by fixing a characteristic peak, e.g. C 1s at a binding energy (BE) of 285 eV in this work. (P. 5, L12-14)
- The author should discuss his results with recent literature results and describe the table in the discussion.
This is a good recommendation. Have done. (P. 7, L1-5 and Table 2)
Comments on the Quality of English Language
There is a need for the author to improve the English in the manuscript.
Have done.
